# Salinity Impacts the Functional *mcrA* and *dsrA* Gene Abundances in Everglades Marshes

**DOI:** 10.3390/microorganisms11051180

**Published:** 2023-04-30

**Authors:** Deidra Jordan, John S. Kominoski, Shelby Servais, DeEtta Mills

**Affiliations:** 1Department of Biological Sciences, Florida International University, Miami, FL 33199, USAjkominos@fiu.edu (J.S.K.);; 2International Forensic Research Institute, Florida International University, Miami, FL 33199, USA; 3Institute of the Environment, Florida International University, Miami, FL 33199, USA

**Keywords:** microbiome, functional potential, methanogens, sulfate reducers, saltwater intrusion, Everglades, NexGen sequencing

## Abstract

Coastal wetlands, such as the Everglades, are increasingly being exposed to stressors that have the potential to modify their existing ecological processes because of global climate change. Their soil microbiomes include a population of organisms important for biogeochemical cycling, but continual stresses can disturb the community’s composition, causing functional changes. The Everglades feature wetlands with varied salinity levels, implying that they contain microbial communities with a variety of salt tolerances and microbial functions. Therefore, tracking the effects of stresses on these populations in freshwater and brackish marshes is critical. The study addressed this by utilizing next generation sequencing (NGS) to construct a baseline soil microbial community. The carbon and sulfur cycles were studied by sequencing a microbial functional gene involved in each process, the *mcrA* and *dsrA* functional genes, respectively. Saline was introduced over two years to observe the taxonomic alterations that occurred after a long-term disturbance such as seawater intrusion. It was observed that saltwater dosing increased sulfite reduction in freshwater peat soils and decreased methylotrophy in brackish peat soils. These findings add to the understanding of microbiomes by demonstrating how changes in soil qualities impact communities both before and after a disturbance such as saltwater intrusion.

## 1. Introduction

The microbial-mediated carbon and sulfur cycles are essential processes in wetland ecosystem functioning [1]. Metagenomic sequencing now allows for correlations of microbial communities to the inherent functional capacity driving biogeochemical cycles. Targeted sequencing using highly conserved 16S rRNA genes to elucidate taxonomy, however, is not capable of identifying various phylogenetic groups linked to specific enzymatic pathways [2]. Often, inferences are made as to the functional capacity of an organism by extrapolating from cataloged taxonomic data shown to have traits encoding for similar functional genes markers [3]. Sequencing these functional biomarkers directly instead of extrapolating from taxonomic data can provide a more realistic and directed functional analysis of habitat functions and the consequences of environmental impacts.

Environmental factors play a significant role in structuring complexities and dynamics of soil microbiomes and the responses of these microbiomes are intrinsically linked to climate drivers. Because of this, it is important to not only characterize the structure of soil microbiomes but also to identify those abiotic drivers, such as soil type, temperature, pH, and other metrics, to explain the microbiome responses [4,5]. As disturbances are introduced into their environment, the community shifts in response, resulting in a change in the microbiome composition and functional capacities. Understanding these dynamics can then assist in possible mitigation or knowledgeable predictions of perturbations that alter the ecological function. Monitoring these factors and predicting the outcomes can be achieved by utilizing machine learning models which can monitor an environment or predict provenance. Recently, human microbiome, soil health and forensic science studies have all acknowledged the utility of machine learning algorithms to investigate microbiome data [6,7,8]. Some studies have shown that analyzing high-dimensional data, such as microbiomes, with random forest algorithms provides high accuracy and that they are less prone to overfitting when compared to other supervised machine learning algorithms [9,10,11]. The main distinction between traditional statistical methods and machine learning algorithms is that instead of inferring the connection between the dataset and the variables, an optimized model is used to predict the outcome from a trained dataset. 

### 1.1. Saltwater Perturbation

Climate perturbations are increasing and understanding how ecosystem functions change, together with the responses of microbial communities through the biogeochemical pathways after these perturbations, must be examined. Several studies have shown that saltwater intrusion can alter the biogeochemical functioning of wetlands [12,13,14,15]. Monitoring the responses of microbial communities across various environments allows for a more holistic picture of the effects disturbances have on ecosystems [16,17].

### 1.2. The Everglades

The Everglades is a coastal wetland increasingly threatened by climate change and damaging anthropogenic water management practices. The various salinity levels across the system produce distinctively diverse aboveground biotic communities within freshwater and brackish marshes [18]. Moreover, it can be expected that the soil biogeochemistry differs as well, leading to disparities in microbial metabolism indicating that the microbial-driven functions may not be sustained due to these perturbations. For example, microorganisms within freshwater environments are more likely to perform methanogenesis while those within brackish or saltwater environments are prone to reducing sulfate. When saltwater is introduced into a freshwater system, this increase in ionic concentration results in sulfate-reducing microbes outcompeting those capable of methanogenesis for metabolic substrates for a more energetically favorable process [19]. Consequently, the carbon and sulfur biogeochemical cycles have been disrupted.

### 1.3. Soil Biota and Their Role in Biogeochemical Cycles

In coastal wetlands, anaerobic soil conditions cause slow decomposition and biogeochemical processing rates, allowing organic matter to build up and soils to store the carbon. Within the first step of the carbon cycle, bacteria and fungi break down organic matter. releasing complex carbon compounds into the soil for storage. Some of that carbon is cycled and eventually released as carbon dioxide, CO_2_, or methane, CH_4_ [20]. Elevated salinity can cause several ecosystem and microbial community modifications stemming from habitat changes in ionic strength, alkalization, and sulfidation [14]. Studies have shown that saltwater decreases the amount of stored soil carbon, suggesting increased microbial respiration rates [21,22,23]. However, investigating the functions of microbes by measuring extracellular enzyme activity has shown both increases and decreases in enzymatic activity exhibiting the sensitivity of these communities to saltwater intrusion [21,23]. 

Previously, it was thought that the only microorganisms contributing to biogeochemical cycles were bacteria and eukarya. However, the expanded usage of molecular biology techniques leading to the discovery of archaea, revealed their critical roles within these cycles [24]. Archaea contribute to the carbon, sulfur, and nitrogen cycles and are abundant in various ecosystems such as soil and marine waters. One pathway where archaea presence dominates is methanogenesis, a pathway within the carbon cycle that produces CH_4_. Methane, a well-known greenhouse gas, is produced from the reduction of a one-carbon compound such as CO_2_. In soil, the CO_2_ along with hydrogen, H, is reduced to CH_4_ under anoxic conditions. Mediated by methanogens, this process is critical in decomposing organic matter [20]. However, the addition of saltwater brings high levels of sulfate, SO_4_^2−^, and sulfate-reducing bacteria can outcompete methanogens for essential metabolic substrates, thus suppressing methanogenesis [25]. Furthermore, a byproduct of sulfate reduction is HS, which slows plant growth and inhibits nitrification, disrupting the nitrogen cycle [14,26]. 

### 1.4. This Study

The objective of this study was to assess the microbial functional capacity within freshwater habitats and brackish marsh within the Everglades. This was characterized through metagenomic sequencing and inferred by the changes in gene marker abundance encoding for enzymes that catalyze reactions from competing biogeochemical cycles—carbon and sulfur. The methyl-coenzyme M reductase subunit alpha (*mcrA*) and dissimilatory sulfite reductase subunit alpha (*dsrA*) genes were sequenced as a representative biomarker for each respective cycle. Shifts in diversity after salinity manipulation experiments and the driving factors impacting the microbial community structure were assessed. It was hypothesized that:

**Hypothesis 1.** *The microbial community’s functional potential in soil microbiomes adapted to brackish environments will not change with salinity dosage experiments compared to those in freshwater environments*.

## 2. Materials and Methods

### 2.1. Sample Preparation for Metagenomic Sequencing

Archived soil samples from freshwater and brackish marshes within the Everglades National Park, FL, USA, were processed [22] and were used for this microbiome study. Briefly, in the Servais et al. (2020) study, 12 polycarbonate chambers were installed along an 80-m-long boardwalk at freshwater and brackish sites within the Everglades Park. Each chamber was 1.4 m in diameter and embedded 30 cm deep into the soil. Six chambers were designated as ambient controls at each location, and six were designated for saltwater treatments. The saltwater chambers were dosed monthly from October 2014 to November 2016, with artificial seawater (Instant Ocean, Spectrum Brands, Blacksburg, VA, USA) mixed with water from the study site [27]. Thirty-three soil cores from the top 0–10-cm layer of soil were used for this subsequent microbiome study. Control soil samples were collected during the initial year of the study (2014) and were not exposed to saltwater pulses—five chambers from the freshwater site (FW_Y0) and five chambers from the brackish water site (BW_Y0). After two years of monthly saltwater pulses, soil samples were collected again in 2016 from each of the experimentally manipulated chambers and from control chambers to determine the temporal changes within the microbial community; five samples from the freshwater control chambers (FW_Y2), six samples from the freshwater chambers exposed to saltwater (FW_Y2_Saline), six samples from the brackish chambers (BW_Y2), and six samples from the brackish chambers exposed to saltwater (BW_Y2_Saline).

### 2.2. DNA Extraction, NGS Library Construction and Sequencing

Metagenomic DNA was extracted from 250 mg of each soil sample using a ZymoBIOMICS DNA Miniprep Kit (Zymo Research, Irvine, CA, USA) according to the manufacturer’s instructions. The DNA was quantified using a Qubit^®^ dsDNA HS Assay Kit on the Qubit^®^ 2.0 Fluorometer (Thermo Fisher Scientific, Waltham, MA, USA). Aliquots were diluted to 5 ng/μL and stored at 4 °C while the remaining stocks were stored at −20 °C until needed. Extracted DNA was amplified with primers that targeted functional genes. Libraries were prepared by first modifying each functional gene primer set to contain Illumina specific adaptor sequences. For the *dsrA* gene: dsrAF-5′-TCGTCGGCAGCGTCAGATGTGTATAAGAGACAG+ACSCACTGGAAGCACG-3′; dsrAR-5′-GTCTCGTGGGCTCGGAGATGTGTATAAGAGACAG+CGGTGMAGYTCRTCCTG-3′ (Sela-Adler et al., 2017). For the *mcrA* gene: mcrF-5′-TCGTCGGCAGCGTCAGATGTGTATAAGAGACAG+GGTGGTGTMGGATTCACACARTAYGCWACAGC; mcrR-5′-GTCTCGTGGGCTCGGAGATGTGTATAAGAGACAG+TTCATTGCRTAGTTWGGRTAGTT-3′ (Morris et al., 2014) [28].

PCR amplifications were performed on a ProFlex PCR System (Applied Biosystems, Foster City, CA, USA) in a 25 μL volume consisting of 1X Phusion Flash High-Fidelity PCR Master Mix (Thermo Fisher Scientific, Waltham, MA, USA), 0.3 μM of each forward and reverse primer, 12.5 ng of DNA and water added to the volume. PCR cycling conditions were: initial denaturation of 98 °C for 10 s, 30 cycles of 98 °C for 10 s, the individual annealing temperature for each gene at 5 s, extension at 72 °C for 15 s, and a final extension at 72 °C for 1 min and were maintained at 4 °C before removal from the thermal cycler. One microliter of the product was visualized on a bioanalyzer trace to verify its size before sequencing. Amplification products were purified using Bulldog Bio Clean NGS SPRI Beads (Bulldog Bio, Inc., Portsmouth, NH, USA) and quantified using the Qubit^®^ 2.0 Fluorometer and the Qubit^®^ dsDNA HS Assay Kit (Thermo Fisher Scientific). The Nextera XT v2 Index Kit (Illumina, Inc., San Diego, CA, USA) was utilized to create the sequencing libraries. Normalized DNA libraries were diluted to 6 pM combined with 15% PhiX Control Spike (Illumina). Sequencing was performed on a MiSeq 2 × 300 platform (Illumina) with a MiSeq v3 reagent cartridge (Illumina).

### 2.3. Functional Gene Amplicon Data Processing and Statistical Analysis

Demultiplexed paired end amplicons reads were first joined together with PANDAseq, the Ribosomal Database Project (RDP) modified version [29]. The assembled sequences were then processed with the RDP Sequence Initial Processor to further remove low-quality sequences; only high-quality assembled sequences were further processed with the FunGene pipeline [30,31] and reference sequences were downloaded from the FunGene Database (http://fungene.cme.msu.edu/, accessed 1 June 2021). After processing, sequences were clustered at an 80% similarity to the reference sequence [32]. Output from the pipeline was imported into the statistical software R Statistical Software v4.1.1 and analyzed with ”RDPutils” and ”Phyloseq” R packages [33,34,35].

### 2.4. Physiochemical Properties of the Soil Microbiomes

Soil samples from Year 2—controls, salt treatments at two sites—were utilized to determine the abiotic drivers of the communities. Measurements of soil physiochemical properties in terms of porewater biogeochemistry, bulk soil and extracellular enzyme activities were obtained from an earlier study [22] and used to assess correlations between the biomarkers and the physicochemical data. A summary of the measurements from the brackish and freshwater sites are included in Appendix A.

### 2.5. Statistical Analyses

Relationships between the soil physiochemical properties and the microbiome datasets were analyzed with distance-based redundancy analysis (dbRDA) and Mantel tests. dbRDA is a form of redundancy analysis (RDA), but it is not constrained by the linearity that is assumed with Euclidean measures. Instead, a dissimilarity matrix is utilized, which is useful for microbiome data and is considered an extension of multiple linear regression which can analyze multiple predictor variables to explain the variance within several response variables at once [36]. In this case, the predictor variables are soil properties, and the response variables are the operational taxonomic units (OTUs). Dissimilarity matrices for dbRDA were constructed using Bray–Curtis dissimilarity. Biplots were produced from dbRDA to determine significant abiotic properties that are the driving factors of the variation within each of the communities [37]. Mantel tests were performed to calculate linear correlations between the OTUs and the soil properties. Each function was performed in R statistical software v4.1.1 with the vegan package [33,38].

## 3. Results

### 3.1. Beta Diversity Patterns of Functional Genes

Beta diversity measurements determined significant differences between the freshwater and brackish sites in each of the functional genes, indicating unique a microbial composition between the two sites. To estimate this dissimilarity in the functional potential of each gene, an ANOSIM test using Bray–Curtis dissimilarity was utilized, Table 1. The test was measured at a *p* < 0.05 level of significance and the sample dissimilarity was visualized using principal coordinates analysis (PcoA) ordination plots (Figure 1 and Figure 2) [39,40]. 

#### 3.1.1. The *mcrA* Gene

The *mcrA* genes differed between the freshwater and brackish sites (see Figure 1). A high similarity was observed within the brackish site over time and even after treatment. This suggests that the brackish water did not vary over time and the saltwater did not influence the overall community functional composition. This is supported by the relative abundance graph (Figure 3), as the brackish water samples—treated or untreated—were dominated by *Methanobacterium* and *Methanotherobacter*. Both taxa are hydrogenotrophic methanogens that co-exist with acetoclastic sulfate reducers that produce hydrogen as metabolic by-products, thus providing the substrates needed without competing for acetate or other intermediates that are utilized by both guilds [41]. But the freshwater site exhibited temporal variation and within the sampled chambers. The separation of data points within the freshwater site suggests that the diversity varied both between chambers and with treatment. However, Year 2 communities did cluster closer together than the samples from the initial collection, Year 0, suggesting that their functional guilds were more similar. Most of the variation within the freshwater chambers occurred due to a shift in abundance of different taxa with *Methanothermobacter* relative abundance increasing and a corresponding decrease in *Methanocella*. *Methanothermobacter* has been shown to tolerate halophilic conditions well as well as heat tolerance—both factors could be in play within the freshwater saline-treated chambers [42].

The taxonomic composition differed greatly between the two sites as well. The freshwater site (63.58 ± 32.77) had a higher diversity than the brackish site (25.5 ± 7.5) as indicated in Figure 3, and by the mean of their Inverse Shannon diversity metric. The brackish site had a relative abundance of 89.8% of *Methanobacterium*, 7.7% of *Methanothermobacter*, and 1.1% of *Methanocella*; the remaining genera were present at less than 1%. The most abundant genera at the freshwater site were *Methanobacterium* (32.9%), *Methanothermobacter* (28.3%), *Methanocella* (7.9%), *Methanothrix* (5%), and *Methanoregula* (4.9%) with remaining taxa present at less than 4.9%. *Methanoregula* was not present at the brackish site but was in the freshwater and decreased from 8.8% to 5.1% after the saltwater treatment, suggesting these taxa may have a lower tolerance for saltwater; thus, the additional salinity affected its ecological functional capacity. *Methanobacterium* also decreased in the freshwater after the salinity treatment, from 31.2% to 26.14% relative abundance.

#### 3.1.2. The *dsrA* Gene

The *dsrA* gene also differed a great deal between the freshwater and brackish sites (see Figure 2). Temporal shifts within both the freshwater and brackish sites were detected. These differences show the natural variation of the communities over time. Salinity pulses significantly affected the *dsrA* gene microbial community composition at the freshwater site as displayed by the distinct groupings in the plot. Salinity had little effect on the community within the brackish site, but the saltwater manipulation experiments caused significant differences within this functional guild at the freshwater site. As observed earlier, the opposite effect occurred with analysis of the *mcrA* gene at this site.

### 3.2. Shifts in Microbial Functional Potential after Salinization

In the *dsrA* gene, over 75% of clusters were classified as uncultured sulfate-reducing bacteria, even at the genus level, Figure 4. All taxonomically identified taxa were present at less than 2% relative abundance. Notable differences between the brackish and freshwater sites included the presence of *Desulfatiglans* in the brackish soil and its absence in the freshwater soil. Additionally, *Carboxydothermus* and *Desulfosudis* were present in the freshwater soil and not in the brackish soil. A total of 33 unique OTUs was found at the freshwater site with 14 found in the control samples and 19 in the treatment samples (Figure 5). The OTUs that were observed in the control samples were in the *Desofundulus* and *Carboxydothermus* genera. Shifts in the freshwater site were identified as decreases in *Desulfosudis* (1.09%) and *Desulfofundulus* (0.43%) after saltwater treatment. At the brackish site, the presence of *Desulfohalovibrio*, and *Desulfosarcina* increased by 0.48% and 0.3% respectively, after saltwater treatment while *Desulfovibrio* (0.15%), *Desulfoglaeba* (0.54%) and *Desufofundulus* (0.17%) all decreased after treatment in the brackish site.

### 3.3. Environmental Drivers of the Microbial Communities

The influence each soil parameter had on the communities was first evaluated with the Mantel test. Samples were separated by site within each group to determine significant factors driving the communities before and after saltwater treatment. Mantel tests were performed, and significant factors were identified at a *p* < 0.05 significant level. The *mcrA* community correlated with pH (r = 0.71, *p* < 0.05) in the brackish site before saltwater treatment. but no significant correlations were identified after the treatment. Additionally, no correlations were found in the *mcrA* community at the freshwater site. Total dissolved nitrogen (r = 0.70, *p* < 0.05) and ammonium (r = 0.78, *p* < 0.001) was significantly correlated to the *dsrA* community at the freshwater site after saltwater treatment. 

The factors contributing most to the variation within these communities before and after saltwater were then assessed with dbRDA. The results were plotted on a principal coordinates analysis (PCoA) plot with the amount of variation explained by each axis, CAP1 and CAP2. All soil properties that were identified as having significantly (*p* < 0.05) contributed to the amount of variance within the samples were plotted as vectors (arrows) in each plot. The direction and length of the arrow represent the direction and strength of the relationship to each axis, CAP1 and CAP2. In the *dsrA* community, sulfide drove the variation in the freshwater site (Figure 6), and alkalinity and ammonium in the brackish site (Figure 7). No correlations were found by Mantel tests at the freshwater site in the *mcrA* community and pH, alkalinity, ammonium, and total dissolved nitrogen were identified as the drivers of this variation (Figure 8). At the brackish site of the *mcrA* community, DOC and alkalinity drove the variation before treatment and sulfate, chlorine, salinity and conductivity drove the variation after treatment (Figure 9).

## 4. Discussion

Microbial communities are suggested to have high levels of intrinsic functional redundancy within the taxonomic composition; therefore, disruptions in biogeochemical cycling are likely to be unaffected by the loss of individual species [43]. However, perturbations in the environment have been known to disrupt the taxonomic structure resulting in the loss of microbes that most efficiently perform these functions, thus ultimately diminishing and disrupting the nutrient cycling within the community [15,19]. Although sequencing the functional genes does not provide information on gene expression, it can provide insights that cannot be obtained with just 16S rRNA genes—the functional potential or capacity of a microbial community to perform critical environmental functions. Results of this study shed light on the differences in composition of the microbial communities involved in the carbon and sulfur cycling and on how they are affected after a perturbation such as saltwater intrusion in two different marshes.

Comparison of the functional microbial community structure between the freshwater and brackish sites supports previous findings where salinity gradients displayed unique compositional patterns [21,44,45]. This could be explained by the capability of a microorganism’s osmoregulatory functions. Many microorganisms do not contain the metabolic capacity to quickly adapt to the additional osmotic stress of saltwater; thus, a significant change in diversity or a shift in the taxonomic composition of a community occurs as many microbes reach their own cellular limitations [46,47]. In this study, the Inverse Simpson index revealed that individual samples from the brackish site seemed less diverse than those from the freshwater site in the *mcrA* functional gene, supporting this claim. Within the *dsrA* functional gene, the brackish site (104.66 ± 62.7) seemed to have an overall higher mean alpha diversity than the freshwater (62.31 ± 32.69). These results can be explained by the increase in sulfate reduction that is required by the larger amount of sulfate present in saltwater environments [19].

Over time the microbial composition shifted within the freshwater site, monitored through the *dsrA* gene. Temporal changes were identified within the brackish site only at the *dsrA* gene. This corresponds with earlier research that showed temporal changes in these samples for total percentage of nitrogen and extracellular enzymes, arylsulfatase and phosphatase [22]. In that study, total percentage of bulk soil nitrogen increased after two years (from 0.8% to 1.2%), and temporal variation occurred in arylsulfatase and phosphatase extracellular enzymes. 

Saltwater manipulation significantly affected the composition of the *dsrA* (R = 0.227, *p* < 0.05) at the freshwater site. An increase in the functional *dsrA* gene diversity and abundance could be suggested by the significant increase in porewater concentrations of sulfate (from 0.01 to 150.03 mg L^−1^) and sulfide (from 0 to 0.08 μmol L^−1^) after saltwater treatment [22]. The current study saw more diversity within this gene marker after saltwater manipulation. This is the opposite of what has been found in a separate study where gene abundance decreased with saltwater manipulation [5]. Within that study, functional genes were measured with GeoChip, a microarray method, and a notable decrease occurred along the salinity gradient at the *dsrA* gene. The perturbation seemed to lead to a shift in composition which likely shifted the functional capacity. Significant increases, as determined by differential abundance analysis, in various OTUs were identified after saltwater manipulation, but limitations in the sequences in archived databases hindered taxonomic classification. Sequences are assigned a taxonomic affiliation by comparing them to the reference sequences present within a database. Therefore, sequences can only be assigned if they reach a specified percentage similarity to a reference within this database. Over 75% of the sequences were assigned an unclassified taxonomic affiliation; therefore, many of these sequences have yet to be classified.

The only community that significantly differed after saltwater manipulation in the brackish site was the *mcrA* community. Methanogenesis is already a limited pathway within more oligohaline environments because of the increased presence of sulfate. Sulfate reducers often outcompete methanogens for the same substrate, hydrogen, or acetate, which may be happening here as the *dsrA* community did not significantly shift at this brackish site, but the *mcrA* community did. Previous studies have suggested that methylotrophic methanogens are likely to be favored in these sulfate-rich environments because they would be less competitive with sulfate reducers [48,49]. The brackish site is still dominated by hydrogenotrophic methanogens before and after saltwater, namely the *Methanobacterium* genus.

A notable decrease in dissolved organic carbon (DOC) was seen after saltwater manipulation, indicating the loss of carbon by some biogeochemical function. DOC has been shown to have a correlation with specific methanogenic archaea, suggesting methanogenesis rates vary dependent upon the level of DOC and composition of taxa [50]. The Liu et al. study found that measurements of methane production were significantly correlated with DOC and that methanobacteriales (hydrogenotrophic methanogens) were found to be predominantly associated with the DOC-rich marsh. In the current study, OTUs from methanobacteriales were found to be more abundant at the control site, which had significantly higher levels of DOC. These results suggest that methanogenesis may be occurring at a higher rate after the saltwater treatment than before. Findings from this study further support the findings of [19] where hydrogenotrophic sulfate reducers and hydrogenotrophic methanogens, which are known competitors, were found to coexist in sulfate-rich environments. However, a separate study investigating Everglades soils after four years of saltwater dosing did not observe a significant shift in DOC or methanogens [51]. This suggests that communities may be more sensitive to short-term inputs and highlights the importance of closely monitoring these communities.

In conclusion, taxonomic increases identified within each of each these functional genes after saltwater treatment suggest an increase in functional potential. Therefore, with more microbes to participate in a specific function, increased rates of activity can be expected. The opposite can be expected if decreases in taxa occurred. Results from this study provided insight into which taxa are more sensitive to changes in saltwater concentrations within each marsh. It was hypothesized that the taxa in the brackish site would not be as affected by the saltwater treatment because the microbes were already accustomed to saline conditions. This was the case for the *dsrA* functional gene, but not *mcrA*, whose community structure shifted after saltwater treatment. This change in taxonomic composition suggests that rates of methanogenesis are expected to shift as saltwater is introduced into the brackish site. These results provide insight for climate scientists because an area of concern is the possibility of increased carbon degradation with saltwater intrusion resulting in higher levels of methane being released.

Assessing factors that shape the community after saltwater manipulation provides insight that could potentially predict how the microbiome will react in the future, especially with the expectation of climate change bringing increased salinity levels into coastal wetlands. Additionally, bioinformatic machine-learning techniques have the potential to aid in determining whether a soil community has been affected by an outside perturbation utilizing its genetic material. 

Various relationships were determined between the microbial communities and each soil parameter before and after saltwater manipulation. In the freshwater site, the addition of saltwater created significant positive correlations in the *dsrA* community. A relationship could not be established in the bacteria community before the saltwater treatment, but several other studies have found relationships between pH and conductivity with their bacterial communities [21,51,52]. Another study concluded that bacteria community structure is shaped by salinity gradients [5]. Results from this study only established relationships in bacteria with DOC and soluble reactive phosphorus (SRP) after saltwater manipulation at both the brackish and the freshwater site. These factors directly affected the community variation because DOC is the main substrate for heterotrophic bacteria which were found to be the most abundant phyla observed within this site. Shifts within the proteobacteria, firmicutes, and actinobacteria, all known heterotrophs, were found to be driven by these substrates [53]. A correlation between salinity was not directly established as a driver of the community variation. 

A significant correlation with pH was found in the *mcrA* brackish site, similar to what was observed in [54]. The authors evaluated the structure of methanogens within a freshwater–brackish marsh and concluded that salinity, together with pH, greatly controlled the community structure. In the current study, soil properties driving the variation within this community differed before and after treatment, further displaying the effect of salinity on the *mcrA* community in a brackish environment. 

Sulfide was seen to influence the microbial community within the *dsrA* gene after long-term saltwater inputs in the freshwater community. This was expected because saltwater brings in sulfate and the sulfate reducers in turn produce sulfide [55]. 

In conclusion, these results show that overall different nutrients can shape the taxonomic composition of the microbial communities before and after saltwater perturbations. The variation in the relationship with the soil properties and the community within different microbial groups suggests a potential for shifts in the community structure and ultimately the ecological function of that community. Knowing that saltwater affects these relationships provides information useful for climate models and for predicting how the microbial functional community may respond.

## Figures and Tables

**Figure 1 microorganisms-11-01180-f001:**
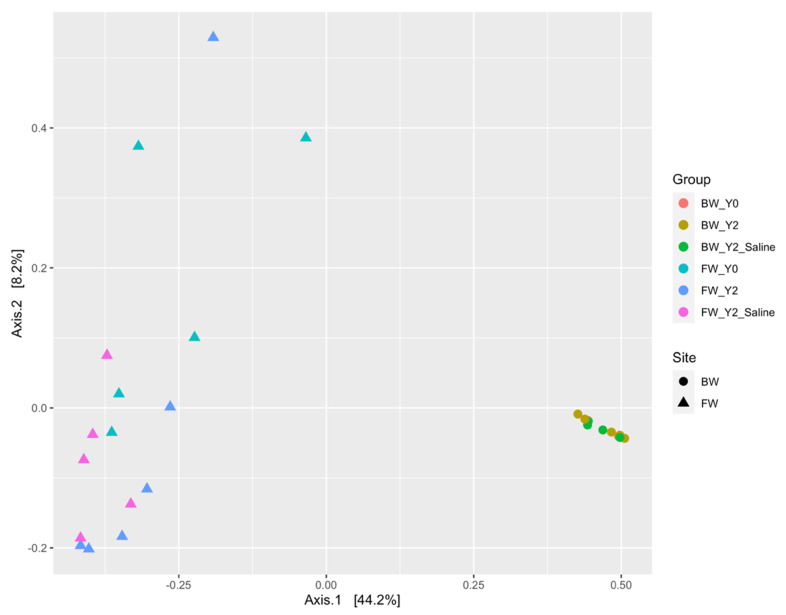
Principal coordinates analysis (PcoA) plot displaying clear separation between brackish (circle) and freshwater (triangle) sites from *mcrA* microbial community profiles. The freshwater site displays high microbial diversity between samples. However, the lack of distinct grouping indicates that the freshwater site experienced no significant changes over time or after treatment. Significant overlap of samples within the brackish site indicates no change over time or after treatment in the *mcrA* microbial community. Each site is defined as: BW = brackish water site, FW = freshwater site. Each group label is as follows: Y is the year, Y0, baseline, time zero; Y2, time year 2 and Saline is the treated mesocosm in each group.

**Figure 2 microorganisms-11-01180-f002:**
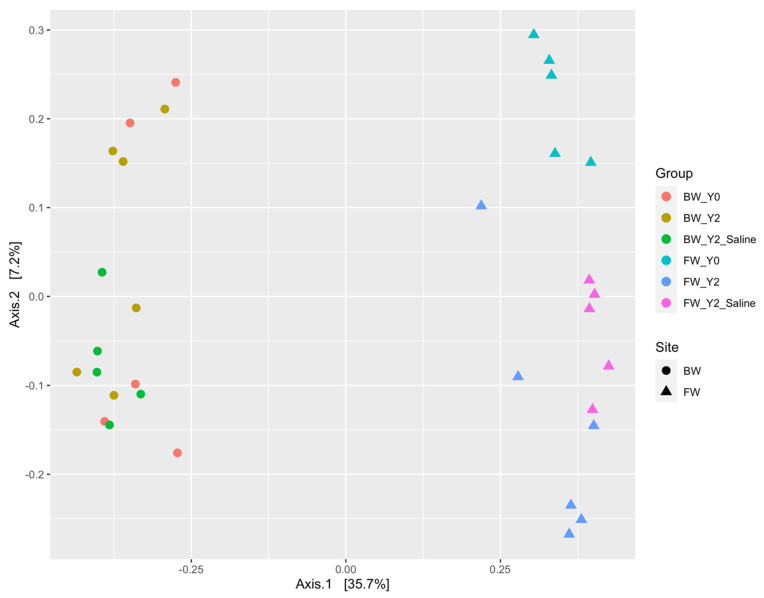
Principal coordinates analysis (PcoA) plot displaying clear separation between brackish (circle) and freshwater (triangle) sites from *dsrA* microbial community profiles. Distinct grouping within the samples from the freshwater site indicates changes in diversity temporally and after saltwater treatment. Overlap with the BW_Y2 and BW_Y2_Saline samples suggests that saltwater treatment did not significantly shift the *dsrA* microbial community. Distinct grouping among the BW_Y0 and BW_Y2 samples suggests temporal changes in the brackish site within the *dsrA* microbial community.

**Figure 3 microorganisms-11-01180-f003:**
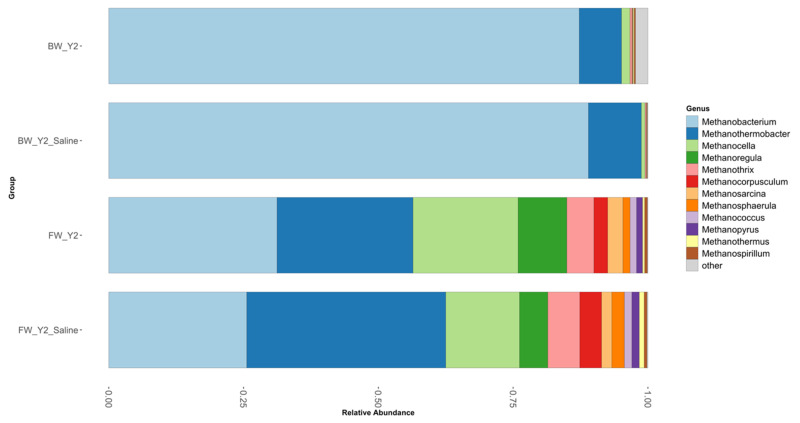
Relative abundance of the most abundant genera found within the Methanobacteria phylum of the *mcrA* microbial community. The brackish (**top** two bars) and freshwater site (**bottom** two bars) displayed differences in observed diversity as exhibited by the taxonomic affiliations.

**Figure 4 microorganisms-11-01180-f004:**
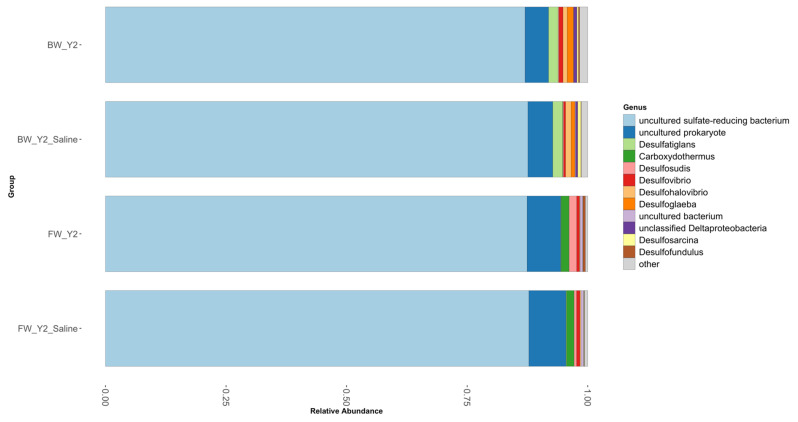
Relative abundance of the most abundant genera found within the *dsrA* microbial community. The brackish (**top** two bars) and freshwater site (**bottom** two bars) both had a unique observed composition.

**Figure 5 microorganisms-11-01180-f005:**
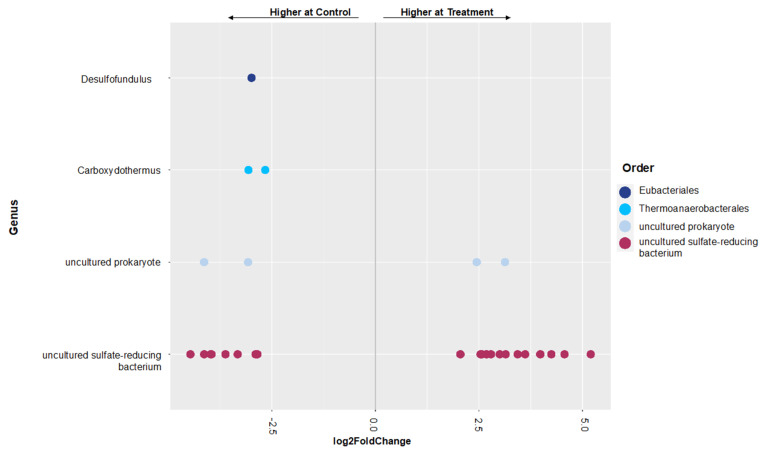
Differential abundance plot between year 2 samples at the freshwater site-control and saltwater treatment. Negative log fold changes represent significant differential abundance at the control samples and positive represent the treatment samples.

**Figure 6 microorganisms-11-01180-f006:**
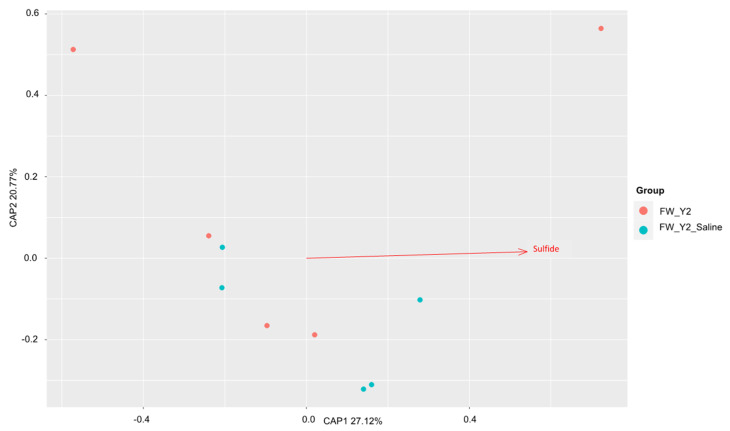
The distance-based redundancy analysis (db-RDA) diagram shows the distribution of soil *dsrA* communities before and after treatment at the freshwater sites. Soil property depicted as red arrow points to positive associations with communities.

**Figure 7 microorganisms-11-01180-f007:**
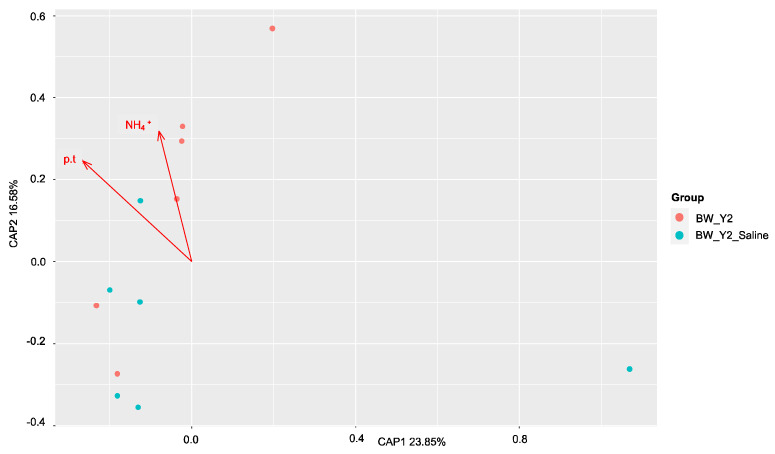
The distance-based redundancy analysis (db-RDA) diagram shows the distribution of soil *dsrA* communities before and after treatment at the brackish sites. Soil properties depicted as red arrows point to positive associations with communities. Phosphorus (p.t), ammonium (NH_4_).

**Figure 8 microorganisms-11-01180-f008:**
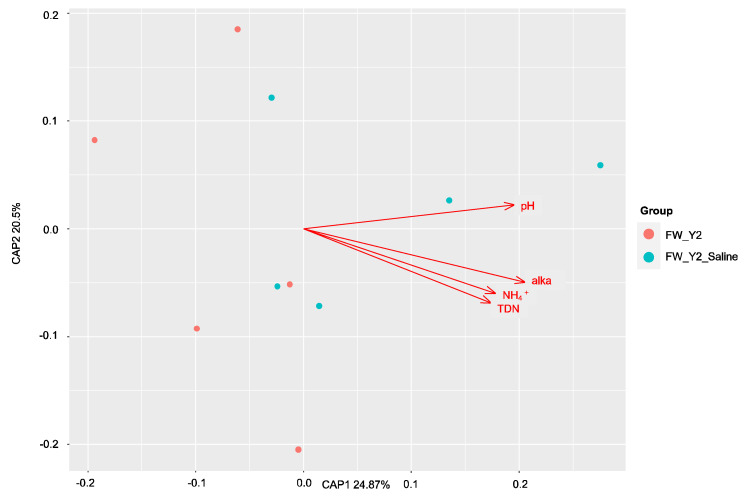
The distance-based redundancy analysis (db-RDA) diagram shows the distribution of soil *mcrA* communities before and after treatment at the freshwater sites. Soil properties depicted as red arrows point to positive associations with communities. Alkalinity (alka), ammonium (NH_4_), total dissolved nitrogen (TDN).

**Figure 9 microorganisms-11-01180-f009:**
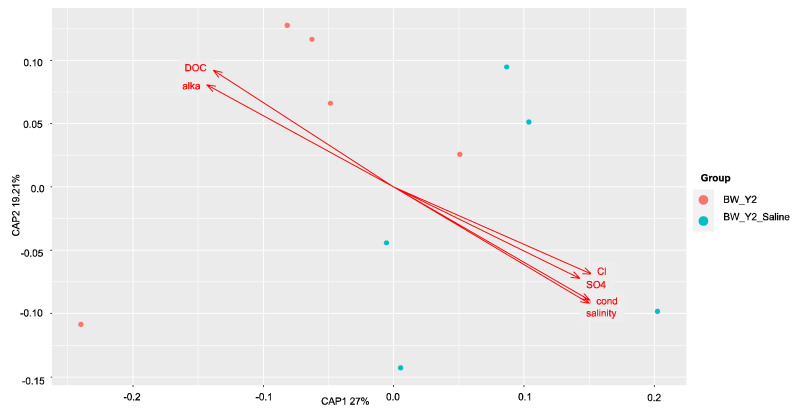
The distance-based redundancy analysis (db-RDA) diagram shows the distribution of soil *mcrA* communities before and after treatment at the brackish sites. Soil properties depicted as red arrows point to positive associations with communities. Dissolved organic carbon (DOC), alkalinity (alka), sulfate (SO_4_), chlorine (Cl), conductivity (cond).

**Table 1 microorganisms-11-01180-t001:** Results of Analysis of Similarities (ANOSIM) test using Bray–Curtis dissimilarity. Statistically significant differences are bolded. R values closer to 1 suggest dissimilarity and negative R values can suggest variability within replicates.

	Temporal		Treatment	
	BW_Y0 vs. BW_Y2	FW_Y0 vs. FW_Y2	BW_Y2 vs. BW_Y2_Saline	FW_Y2 vs. FW_Y2_Saline
** *Gene* **				
** *mcrA* **	R = 0.291, *p* > 0.05	R = 0.151, *p* > 0.05	**R = 0.319, *p* = 0.051**	R = 0.093, *p* > 0.05
** *dsrA* **	**R = 0.307, *p* < 0.05**	**R = 0.608, *p* < 0.05**	R = -0.045, *p* > 0.05	**R = 0.227, *p* < 0.05**

## Data Availability

Data will be made available by the authors upon request.

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
