# Peer review of "Salinity Impacts the Functional *mcrA* and *dsrA* Gene Abundances in Everglades Marshes"

_microorganisms, 2023, doi:10.3390/microorganisms11051180_

Round 1

Reviewer 1 Report

The manuscript is a continuation of an earlier work done in the Everglades (Effects of saltwater pulses on soil microbial enzymes and organic matter breakdown in fresh water and brackish coastal wetlands). This work, on the other end, investigates various aspects, in the same sites previously analyzed, of microbial ecology. In particular, the amplification in the different samples of the mcrA genes, as representative of the carbon cycle, and dsrA genes for the sulfur cycle, in freshwater and brackish water environments, with and without an additional salinization over the course of two years, is reported.

The work is well written and the analysis is well set up. The methodology used is appropriate and the data processing is thorough and interesting.

The results obtained are quite interesting and show the perturbation on the microbial communities of the salinization of the environment and report possible implications in climate change.

In particular, I suggest reporting the meaning of the abbreviations of the sampling sites at least in the caption of figure 1.

On line 422 report full SRP.

Enlish language is fine

Author Response

Thank you for your review. We added the explanation of the legend for figure 1 in the figure legend. We also added soluble reactive phosphorus  to define the SRP.

Reviewer 2 Report

The revised manuscript studies marshes microbiomes in the function of environmental factors. This is interesting as microbial composition depends strongly on environmental conditions being related to the climate. Using NGS and modern computing techniques allows for better understanding both microbial community structure and the processes affecting it on genetic level. For these I found the work interesting and worth being published in Microorganisms Journal.

The work has interesting and well organized introduction leading to its aim.

Experiment is well designed and study conducted too. I only found that Table 1 is missing.

Results are clearly described followed by standard graphs of a standard quality. Authors could try to increase font size on them to make figures more readable.

Discussion explains the observed outcome and leads to conclusions.

Author Response

Thank you for your review. I deleted the reference to table 1 and just added the actual sequences to the paragraph. I also realized the other table 2 was missing in the copy I have so I updated and added that table (now table 1). I was not able to increase the font size of the figures as it was a limitation of the R software module that was used.
